# Immunocapture Magnetic Beads Enhanced the LAMP-CRISPR/Cas12a Method for the Sensitive, Specific, and Visual Detection of *Campylobacter jejuni*

**DOI:** 10.3390/bios12030154

**Published:** 2022-03-02

**Authors:** Chao Li, Xuan Chen, Renqiao Wen, Peng Ma, Kui Gu, Cui Li, Changyu Zhou, Changwei Lei, Yizhi Tang, Hongning Wang

**Affiliations:** 1Animal Disease Prevention and Food Safety Key Laboratory of Sichuan Province, College of Life Sciences, Sichuan University, Chengdu 610064, China; lichaomeet@gmail.com (C.L.); 2019222040101@stu.scu.edu.cn (X.C.); wenrenqiao132@163.com (R.W.); ma_peng99@163.com (P.M.); gukui0404@stu.scu.edu.cn (K.G.); cui919@126.com (C.L.); l1z2l3y4@163.com (C.Z.); leichangwei@scu.edu.cn (C.L.); 2Key Laboratory of Bio-Resource and Eco-Environment of Ministry of Education, College of Life Sciences, Sichuan University, Chengdu 610064, China

**Keywords:** *Campylobacter jejuni*, food-borne pathogens, ICB-LAMP-CRISPR/Cas12a, point of care, visual detection

## Abstract

*Campylobacter jejuni* is one of the most important causes of food-borne infectious disease, and poses challenges to food safety and public health. Establishing a rapid, accurate, sensitive, and simple detection method for *C. jejuni* enables early diagnosis, early intervention, and prevention of pathogen transmission. In this study, an immunocapture magnetic bead (ICB)-enhanced loop-mediated isothermal amplification (LAMP) CRISPR/Cas12a method (ICB-LAMP-CRISPR/Cas12a) was developed for the rapid and visual detection of *C. jejuni*. Using the ICB-LAMP-CRISPR/Cas12a method, *C. jejuni* was first captured by ICB, and the bacterial genomic DNA was then released by heating and used in the LAMP reaction. After the LAMP reaction, LAMP products were mixed and detected by the CRISPR/Cas12a cleavage mixture. This ICB-LAMP-CRISPR/Cas12a method could detect a minimum of 8 CFU/mL of *C. jejuni* within 70 min. Additionally, the method was performed in a closed tube in addition to ICB capture, which eliminates the need to separate preamplification and transfer of amplified products to avoid aerosol pollution. The ICB-LAMP-CRISPR/Cas12a method was further validated by testing 31 *C. jejuni*-positive fecal samples from different layer farms. This method is an all-in-one, simple, rapid, ultrasensitive, ultraspecific, visual detection method for instrument-free diagnosis of *C. jejuni*, and has wide application potential in future work.

## 1. Introduction

*Campylobacter jejuni* (*C. jejuni*) is a foodborne bacterial pathogen that causes gastroenteritis in humans [1]. As a zoonotic pathogen, *C. jejuni* is widely distributed in food animal species, especially chickens, and it is transmitted to humans primarily through the foodborne route [2,3]. *C. jejuni* infection generally causes self-limited diarrhea in humans; however, it may induce severe or systemic infections in immunocompromised or young and elderly patients [3]. Approximately 550 million people suffer from diarrhea every year, including 220 million children under the age of five years old [3]. In the U.S., it is estimated that *Campylobacter* is responsible for more than 1.3 million cases of illnesses each year [4]. In England, the total cost of *Campylobacter* species infection was estimated to be 116 million dollars each year [5]. Therefore, rapid and accurate detection of foodborne pathogens, such as *C. jejuni*, could help to improve public health and increase food safety.

National (GB 4789.9-2014) and international (ISO 10272-1-2017) standards for the detection of *C. jejuni* include conventional enrichment culture and biochemical identification methods [6,7,8]. However, these methods are time-consuming and expensive, with high rates of false negatives. Thus, a rapid and accurate detection method for *C. jejuni* would be an important improvement to current standards for human food safety.

Molecular diagnostic methods targeting nucleic acids, such as polymerase chain reaction (PCR), quantitative polymerase chain reaction (qPCR), droplet digital PCR (ddPCR), next-generation sequencing (NGS), and loop-mediated isothermal amplification (LAMP), are often used in pathogen screening, and are presented in a large number of industry and local standards (http://www.cssn.net.cn/cssn/front/listpage.jsp (accessed on 1 October 2021)) [9,10,11,12,13]. However, PCR, qPCR, ddPCR, and other alternating temperature, amplification-based detection techniques require trained personnel, expensive equipment, and long reaction times. These requirements make them unsuitable for simple, fast, and point-of-care (POC) molecular diagnosis [14,15,16]. NGS technology has emerged as a promising approach for pathogen detection due to its high-throughput and superior sensitivity and specificity [17]. However, the sequencing costs, the time associated with NGS, and the requirement for professional bioinformatics personnel limit its widespread application [18]. In contrast, LAMP-based isothermal amplification technology (IAT) is a possible substitute for PCR [19]. Compared to other alternating temperature amplification technologies, IAT has the advantages of being rapid, efficient, specific, and equipment-free, and is now considered to be comparable to PCR [20]. However, it is challenging to use IAT for accurate and reliable POC testing due to undesired amplification signals, which can lead to false-positive results [21]. Additionally, the resulting output of the IAT requires considerable improvement before practical use is possible [22,23]. On the basis of IAT, further improving accuracy and reducing false positives are conducive to ensuring accurate detection.

CRISPR/Cas (clustered regularly interspaced short palindromic repeats/CRISPR associated)-based methods, which have the properties of being ultrasensitive and portable for diagnostic tests, may revolutionize methods for nucleic acid detection [24]. The CRISPR/Cas detection system is composed of Cas endonucleases and a programmable single guide RNA (sgRNA). sgRNAs are single-stranded RNAs that complement to a specific target sequence, which guides Cas endonucleases to cleave target fragments specifically [25,26]. The target of CRISPR/Cas detection systems varies by the type of Cas endonucleases used in a given system. The most commonly used Cas endonucleases are Cas9, Cas12a, Cas13a, and Cas14 [27]. Cas9 and Cas12a target double-stranded DNA, Cas13a targets RNA, and Cas14 targets single-stranded DNA.

The CRISPR/Cas system has shown great promise for the development of next-generation POC molecular diagnostic technology due to its high specificity and reliability [28]. The combination of a CRISPR/Cas detection system and IAT may thus improve the detection sensitivity and specificity of a POC nucleic acid detection system. Chen et al. established a DETECTR detection system based on CRISPR/Cas12 for the detection of SARS-CoV-2 [29]. Gootenberg et al. developed the virus detection technology SHERLOCK based on CRISPR/Cas13, and applied SHERLOCK to detect SARS-CoV-2 [30,31]. Li et al. developed a Cas12a-based HOLMES method for the rapid detection of target DNA and RNA within 1 h [32]. Although these methods had high specificity and sensitivity, they could still be optimized methodologically to improve the specificity and sensitivity. Joung et al. modified the SHERLOCK detection method by adding magnetic beads to enrich the RNA in samples during the process of sample preparation [33]. The sensitivity was further improved by increasing the number of initial templates for nucleic acid amplification [33].

In this study, we report a method for the detection of *C. jejuni* that integrates immunocapture magnetic beads (ICB), LAMP, and CRISPR/Cas12a (ICB-LAMP-CRISPR/Cas12a) to enhance the sensitivity and specificity of this method. This method is an all-in-one, simple, rapid, ultrasensitive, and ultraspecific visual detection method for instrument-free diagnosis of *C. jejuni*.

## 2. Materials and Methods

### 2.1. Materials

NHS-magnetic beads and a magnetic separator were purchased from BEAVER (Suzhou, China). Skirrow selective medium was purchased from Beijing Land Bridge Technology Co., Ltd. (Beijing, China). Brucella medium, Luria broth medium, and *C. jejuni* polyclonal antibodies were purchased from Thermo Fisher Scientific, Inc. (Shanghai, China). Phosphate buffer saline (PBS) and mineral oil were purchased from Sangon Biotech Co., Ltd. (Shanghai, China). Bst 3.0 DNA polymerase and EnGen^®^ Lba Cas12a (Cpf1) were purchased from New England Biolabs Inc (Ipswich, MA, USA). RNase inhibitor and T-green transilluminator were purchased from TIANGEN Biotech Co., Ltd. (Beijing, China). A dry bath incubator was purchased from Hangzhou Miu Instruments Co., Ltd. (Hangzhou, China). Primers, ssDNA-FQ probes, and sgRNAs were synthesized by Tsingke Biotechnology Co., Ltd. (Beijing, China). Taq premix for PCR and qPCR premix were purchased from TransGen Biotech Co., Ltd. (Beijing, China). The CFX96 Touch Real-Time PCR Detection System was purchased from Bio-Rad Laboratories, Inc. (Hercules, CA, USA). A UVP UVsolo Touch was purchased from Analytik Jena AG (Jena, Germany). All the tested strains, including *C. jejuni* NCTC 11168, *Campylobacter coli*, *Escherichia coli*, *Shigella flexneri*, *Klebsiella pneumoniae*, *Proteus mirabilis*, *S. enteritidis*, and 31 *C. jejuni*-positive fecal samples from different layer farms were stored in our laboratory. *C. jejuni NCTC 11168* and *Campylobacter coli* were cultured in Brucella medium using the following growth conditions (42 °C, 5% O_2_, 10% CO_2_, and 85% N_2_, 44–48 h). The other non-*C. jejuni* were cultured in Luria broth medium (37 °C for 24 h).

### 2.2. Preparation and Evaluation of ICB

ICB were prepared according to the manufacturer’s instructions of the BeaverBeads™ Mag NHS Kit. Briefly, 500 μL of NHS-magnetic beads (10 mg/mL) was mixed with 500 μL of *C. jejuni* polyclonal antibody (0.6 mg/mL) and coated at room temperature for 2 h in a shaking incubator (80 rpm). After magnetic separation, the remainder of the antibody in the tubes was removed and washed three times with PBS. Then, the ICB was resuspended in PBS and stored at 4 °C until further use. The *C. jejuni* capture time of the ICB was determined by the optical density (OD) changes. One milliliters of *C. jejuni* (8 × 10^3^ CFU/mL) was mixed with ten milliliters of ICB and incubated at room temperature for 60 min in a shaking incubator (80 rpm). The OD in the supernatant was measured every 10 min after magnetic separation. The *C. jejuni* capture efficiency of ICB was determined by gradient dilutions and plate counting. *C. jejuni* bacterial solutions (8 × 10^1^–8 × 10^3^ CFU/mL) were mixed with 10 μL of ICB. After incubation and magnetic separation, the remaining liquid was coated on Brucella medium solid plates and incubated for 44 h (42 °C, 5% O_2_, 10% CO_2_, and 85% N_2_). Plate counts were performed to evaluate the capture efficiency of ICB.

### 2.3. Primers, sgRNAs, and ssDNA Probe Design

The *hipO* (*hippurate hydrolase*) gene (NC_002163.1:c919731-918580) is a conserved gene in *C. jejuni*, and is commonly used for *C. jejuni* identification [34,35]. The LAMP primers (outer primers: F3, B3; inner primers: FIP, BIP) and sgRNAs used in this study were designed based on the *hipO* gene. These LAMP primers were designed using Primer Explorer V5 software (http://primerexplorer.jp/lampv5/index.html (accessed on 1 October 2021)), and sgRNAs were designed in CHOPCHOP (http://chopchop.cbu.uib.no/ (accessed on 1 October 2021)). The outer primers for LAMP can be used in PCR and qPCR amplification. The inner primer FIP was composed of F2 and F1c, while BIP was composed of B2 and B1c. The ssDNA probe was determined based on the Cas endonucleases used, which was a six-nucleotide (nt) single-stranded DNA (5′-TTTTTT-3′) labeled by 5′ 6-FAM (Fluorescein) and a 3′ Iowa Black FQ quencher. The sequence information of all primers, sgRNAs, and ssDNA probes used are listed in Appendix A.

### 2.4. LAMP/PCR/qPCR Amplification Reaction

The optimized LAMP reaction system contained 1 μL of isothermal amplification buffer II, 6 mM Mg^2+^, 320 U/mL of Bst 3.0 DNA polymerase, 1.2 mM of deoxyribonucleotide (dNTPs), 0.2 μM of outer primer (F3/B3), 1.6 μM of inner primer (FIP/BIP), genomic DNA (2 μL for routine LAMP, and 5 μL for ICB-LAMP), and nuclease-free water up to 10 μL. This system was incubated at 65 °C for 30 min. To reduce heat transfer and prevent contamination, this 10 μL LAMP reaction system was covered with 20 μL of mineral oil. For PCR amplification, the 25 μL PCR mixtures were composed of 1 μL of forward and reverse primers (0.4 μM), 12.5 μL of Taq premix, 2 μL of template DNA, and nuclease-free water to the final reaction volume. These reaction mixtures were then incubated in a thermocycler using a three-step PCR protocol: 95 °C, 10 min for predenaturation; 95 °C, 10 s for denaturation; 54 °C, 10 s for annealing; 72 °C, 30 s for elongation; 30 cycles, and 72 °C, 10 min for final elongation. Both LAMP and PCR products can be verified using gel electrophoresis on a 3% agarose gel at 120 V for 30 min, and can be visualized using a gel image analysis system (UVP UVsolo Touch, Jena, Germany). For qPCR amplification, 20 μL of the qPCR mixture was composed of 0.4 μL of forward and reverse primers (0.2 μM), 10 μL of premix, 2 μL template of DNA, and nuclease-free water to the final reaction volume. The reaction mixture was incubated in a CFX96 Touch Real-Time PCR Detection System using a three-step qPCR protocol over 40 cycles: 95 °C, 30 s for denaturation; 95 °C, 5 s for denaturation; 54 °C, 15 s for annealing; and 72 °C, 10 s for elongation. After amplification, melt curve analysis was performed using the default program.

### 2.5. Formation of the ICB-LAMP-CRISPR/Cas12a Detection System

Traditionally, nucleic acid amplification products are often separated and identified by agarose gel electrophoresis. However, it is time-consuming, and non-specific visual results may be caused by nucleic acid dyes [36]. A specific CRISPR/Cas12a cleavage reaction was introduced into the ICB-LAMP system (Figure 1A), which improved the specificity of detection and improved the reading method for the results. The CRISPR/Cas12a cleavage reaction was composed of three key components: Cas12a, sgRNA, and a ssDNA-FQ probe. Cas12a acted as a molecular scissors, which could break the target DNA using its cis-cleavage activity and non-target DNA by trans-cleavage activity [24]. The cis-cleavage activity of Cas12a was guided by the sgRNA, which was designed to specifically recognize target DNA. A ssDNA-FQ probe (Appendix A) was used as non-target DNA, and could be interrupted by the trans-cleavage activity of Cas12a. Due to the different operating temperatures, this LAMP-CRISPR/Cas12a method was divided into 2 parts: LAMP and CRISPR/Cas12a. The LAMP reaction system was placed in the bottom of the tube and incubated at 65 °C for 30 min. The CRISPR/Cas12a cleavage system was placed in the lid of the tube and mixed with the LAMP system once LAMP was finished, and then the reaction occurred at 37 °C for 10 min. The fluorescent results were visually observed immediately under LED blue light. To optimize the CRISPR/Cas12a cleavage system, each reaction was performed at 37 °C for 30 min. The final fluorescent results were visually observed under LED blue light, and real-time monitoring was performed by using a CFX96 Touch Real-Time PCR Detection System.

To test the ICB-LAMP-CRISPR/Cas12a detection system, 1 mL of *C. jejuni* (8 × 10^3^ CFU/mL) was mixed with 10 μL of ICB and incubated at room temperature for 20 min. Then, the captured *C. jejuni* was resuspended in 5 μL of ddH_2_O for template DNA release at 100 °C for 10 min. All the template DNA was used as a template in the LAMP reaction, which was placed at the bottom of the tube. The 10 μL LAMP products were used as the substrate for the CRISPR/Cas12a reaction system, which was added to the lid, and contained 2 μL of NEB 2.1 reaction buffer (10×), 166.67 nM of EnGen Lba Cas12a, 1.5 μM of sgRNA, 1.67 μM of ssDNA-FQ probe, 4 U/μL of RNase inhibitor, and up to 20 μL of nuclease-free water. After 30 min of LAMP amplification, the CRISPR/Cas12a cleavage system was mixed with the LAMP products by shaking. Then, the CRISPR/Cas12a cleavage reaction was conducted at 37 °C for 10 min, and fluorescence results were visually observed immediately under LED blue light.

### 2.6. Specificity and Sensitivity Evaluation of ICB-LAMP-CRISPR/Cas12a

To verify the specificity of ICB-LAMP-CRISPR/Cas12a method, *C. jejuni* NCTC 11168 was used as a positive control strain, 6 non-*C. jejuni* (*C. coli*, *E. coli*, *S. flexneri*, *K. pneumoniae*, *P. mirabilis*, *S. enteritidis*) were used as negative controls, and ddH_2_O was used as a no-template control. Cultures of the test strains (1 mL) were incubated with 10 μL of ICB at room temperature for 20 min. After magnetic separation, the remaining ICB was resuspended in 5 μL of ddH_2_O, and the template DNA was released by boiling at 100 °C for 10 min. Five microliters of template DNA was added to the LAMP reaction and incubated at 65 °C for 30 min, mixed with the CRISPR/Cas12a cleavage system and incubated at 37 °C for 10 min, after which the fluorescence results were visually observed under LED blue light. To verify the sensitivity of ICB-LAMP-CRISPR/Cas12a method, serial tenfold dilutions of *C. jejuni* (8 × 10^1^–8 × 10^3^ CFU/mL) were incubated with 10 μL of ICB at room temperature for 20 min. After magnetic separation, the remaining operations were the same as those used in the specificity evaluation.

## 3. Results

### 3.1. Construction of the ICB-LAMP Reaction System

The ICB-LAMP reaction system was composed of ICB capture and LAMP reactions. Because the NHS-magnetic beads were coated with *C. jejuni* polyclonal antibody (Figure 1A), *C. jejuni* was captured by ICB and then resuspended in 5 μL of ddH_2_O, and the template DNA was released at 100 °C for 10 min after magnetic separation. Approximately 5 μL gDNA was added to the LAMP reaction, and this reaction mix was incubated for 30 min at 65 °C in a dry bath incubator.

To determine the capture ability of ICB, the binding efficiency and capture time were evaluated. The OD600 change was negatively correlated with the incubation time and no significant difference was observed after 20 min of incubation (Figure 2A). Hence, the capture time of ICB was 20 min. Three original bacterial solutions with 8 × 10^1^–8 × 10^3^ CFU/mL were used to determine the binding efficiency. After 20 min of incubation and magnetic separation, the remaining liquid was coated on Brucella medium solid plates (100 μL/culture dish, three repetitions); these plates were incubated for 44 h at 42 °C (5% O_2_, 10% CO_2_, and 85% N_2_). Plate counting was used to evaluate the average binding efficiency of ICB. Appendix A shows that the binding efficiency of ICB depended on the concentration of the bacterial solution used. The lower the bacterial concentration, the higher the capture efficiency. The average binding efficiency of ICB was 92.4%, indicating that the prepared ICB could capture 92.4% of *C. jejuni* in solution within 20 min.

To obtain a better LAMP reaction, the LAMP components, including primers, Mg^2+^, and dNTPs, were optimized. Five groups of LAMP primers (Primers 1–5) based on the *hipO* gene were designed and tested (Appendix A) to determine the optimal primers. As shown in Figure 2B, primer 2 was confirmed for use in the following tests, and the primer binding sites are shown in Figure 2C. Mg^2+^ was optimized from 2 μM to 12 mM, and 6 μM was chosen as the final concentration (Appendix A). The levels of dNTPs in the LAMP reaction were then optimized from 1 mM to 2 mM, and 1.4 mM was selected as the final concentration (Appendix A).

### 3.2. Establishment of the ICB-LAMP-CRISPR/Cas12a Method

Six sgRNAs were designed based on the target DNA (Appendix A), and only sgRNA 4 worked well in the standard program, producing strong fluorescence under LED blue light (Figure 3A). The binding site is shown in Figure 2C. The real-time monitor in the CFX96 Touch Real-Time PCR Detection System also showed that the endpoint fluorescence was highest for sgRNA 4, and this reaction reached its fluorescent threshold (1000) in less than 4 min (Figure 3B). To systematically evaluate the ICB-LAMP-CRISPR/Cas12a method, we tested eight reaction systems (reactions 1–8) with various components (Figure 4). The LAMP products of *C. jejuni* were used as the target sequence. After incubation at 37 °C for 30 min, only reaction 4, containing the target nucleic acid sequence, sgRNA, Cas12a, and the ssDNA-FQ reporter, produced a superbright fluorescent signal that could be directly visualized under LED blue light. In addition, in the real-time fluorescence monitoring experiments, only reaction 4 showed a significantly increased fluorescence signal and a short time to fluorescent threshold (1000). To improve the reaction performance and reduce the reaction volume, the concentrations of sgRNA, Cas12a, and the ssDNA-FQ probe, as well as the substrate volume, were optimized to obtain high cleavage activity. Appendix A shows that the substrate volume (Appendix A) was optimized from 5 μL to 25 μL, the sgRNA concentration (Appendix A) was optimized from 0.67 μM to 1.83 μM, the Cas12a concentration (Appendix A) was optimized from 50 nM to 166.67 nM, and the ssDNA-FQ probe concentration (Appendix A) was optimized from 0.5 μM to 1.67 μM. Finally, a 10 μL substrate volume, 1.83 μM of sgRNA, 166.67 nM of Cas12a, and a 1.67 μM of ssDNA probe were used in the follow-up studies. These results showed that the ICB-LAMP-CRISPR method provides a rapid, one-container approach for the detection of target-specific nucleic acids.

### 3.3. Specificity Evaluation of ICB-LAMP-CRISPR/Cas12a

The specificity of ICB-LAMP-CRISPR/Cas12a was compared with PCR, qPCR, and LAMP. *C. jejuni* and six non-*C. jejuni* (*C. coli*, *E. coli*, *S. flexneri*, *K. pneumoniae*, *P. mirabilis*, and *S. enteritidis*) were incubated with ICB for 20 min, and then resuspended in 5 μL of ddH_2_O after magnetic separation. The template DNA of these tested strains was used in the specificity evaluation. Figure 5 shows that *C. jejuni* could be identified by PCR (Figure 5A), qPCR (Figure 5B), LAMP (Figure 5C), and ICB-LAMP-CRISPR/Cas12a (Figure 5D), without false positives or cross reactions to non-*C. jejuni*. The ICB-LAMP-CRISPR/Cas12a method appeared to possess the same specificity as PCR, qPCR, and LAMP.

### 3.4. Sensitivity and Time Evaluation of ICB-LAMP-CRISPR/Cas12a

In the ICB-LAMP-CRISPR/Cas12a method, all ICB capture, LAMP, and the trans-cleavage activity of Cas12a contributed to signal amplification (Figure 1C). In the specificity evaluation, the sensitivity of the ICB-LAMP-CRISPR/Cas12a method was compared to PCR, qPCR, LAMP, and ICB-LAMP. Serial tenfold dilutions of *C. jejuni* (8 × 10^0^–8 × 10^10^ CFU/mL) were used for sensitivity evaluation. Figure 6 shows that the limit of detection (LOD) of PCR (Figure 6A) was 8 × 10^3^ CFU/mL, the LOD of qPCR (Figure 6B) was 8 × 10^2^ CFU/mL, the LOD of LAMP (Figure 6C) was 8 × 10^3^ CFU/mL, and the LOD of ICB-LAMP (Figure 6D) and ICB-LAMP-CRISPR/Cas12a (Figure 6E) was 8 × 10^0^ CFU/mL. These results were confirmed by agarose gel electrophoresis and real-time monitoring. The ICB-LAMP-CRISPR/Cas12a method appeared to possess extremely high sensitivity compared with PCR, qPCR, and LAMP.

To confirm the time of fluorescent signal production, the cleavage procedure of CRISPR/Cas12a was monitored using a CFX96 Touch Real-Time PCR Detection System and real-time photograph detection. A 1 mL sample of *C. jejuni* (8 × 10^3^ CFU/mL) was detected via the ICB-LAMP-CRISPR/Cas12a method. A visible fluorescence signal was produced at 3 min, and then the fluorescence signal became stronger with time (Figure 7). To cover as many unexpected conditions as possible (especially low concentration samples), increase the error-tolerance rate, and maximize the cutting effect of Cas12a, 10 min was chosen in the following tests.

### 3.5. C. jejuni-Positive Fecal Sample Detection by ICB-LAMP-CRISPR/Cas12a

After a series of experimental optimizations, the finalized conditions of the ICB-LAMP-CRISPR/Cas12a method were obtained. The *C. jejuni*-positive fecal samples (n = 31) from different layer farms were used for the actual sample evaluation of ICB-LAMP-CRISPR/Cas12a. One hundred milligrams of fecal sample were resuspended in 1 mL of PBS, which was then captured by ICB for 20 min and resuspended in 5 μL of ddH_2_O. After magnetic separation, the template DNA was released at 100 °C for 10 min. The template DNA was used as a template for the LAMP reaction, which was incubated at 65 °C for 30 min and then mixed with the CRISPR/Cas12a system at 37 °C for 10 min. All operations could be completed in any environment, with the use of heat blocks, a magnetic separator, pipettes, tips, and a T-green transilluminator (Figure 1D). All results were visible under LED blue light. All 31 *C. jejuni*-positive fecal samples were detected by ICB-LAMP-CRISPR/Cas12a in 70 min (Figure 8). These results illustrate the simplicity and versatility of the ICB-LAMP-CRISPR/Cas12a method.

## 4. Discussion

*C. jejuni* is often associated with food contamination [37,38,39]. Over the past decade, infections caused by *C. jejuni* have been reported in many regions, including Europe, North America, Australia, the Middle East, Africa, and Asia, and their incidence rates have been increasing [40]. With the increase in *C. jejuni* resistance, the infection caused by *C. jejuni* becoming harder to treat [41]. Therefore, the development of a rapid and sensitive diagnostic method to detect and reduce the spread of *C. jejuni* is important for public health. For pathogen detection and early warning, the quick and direct detection of low levels of pathogens is required.

A novel detection method (ICB-LAMP-CRISPR/Cas12a) was constructed, which improved the sensitivity and specificity in three different aspects. The ICB-LAMP-CRISPR/Cas12a method developed herein could specifically recognize and capture free *C. jejuni* in fecal samples within 20 min using the *C. jejuni* antibody-coated magnetic beads. This improved the sensitivity and specificity from the source of the detection method (Figure 1) [42]. Conventional nucleic acid amplification methods require at least 1 × 10^3^ CFU/mL of colonies as a template to perform normal amplification [43,44], while sample enrichment by ICB in the ICB-LAMP-CRISPR/Cas12a method could reach a detection limit of 8 × 10^0^ CFU/mL. For the captured *C. jejuni*, template DNA was released by direct heating cleavage and used for the amplification of target DNA. LAMP primers ensure the specificity, and the efficient Bst 3.0 enzyme ensures the sensitivity. LAMP-amplified target DNA was mixed with the CRISPR/Cas12a system for specific cleavage and efficient fluorescence production. Cas12a endonuclease cut the target DNA under the guidance of a specific sgRNA, and stimulated the trans-cleavage activity to cut the ssDNA-FQ probe to generate fluorescence for signal amplification [45]. The whole detection process was composed of four parts: ICB capture for 20 min, gDNA release for 10 min, LAMP amplification for 30 min, and CRISPR/Cas12a digestion for 10 min, for a total time of 70 min. The temperatures required were 100 °C for gDNA release, 65 °C for LAMP amplification, and 37 °C for CRISPR/Cas12a digestion. Due to the difference between LAMP and Cas12a endonuclease working temperatures, the LAMP amplification system was placed in the bottom of the tube and the CRISPR/Cas12a system was placed in the tube lid. When the LAMP reaction was completed, the CRISPR/Cas12a system was mixed by inversion. As the ICB-LAMP-CRISPR/Cas12a method was completed in a stepwise manner, there was no need for separate and complex manual operations. These results could be visualized under an LED light, thereby significantly simplifying the detection process and eliminating the need for separate lateral flow-based detection [46,47]. Compared to traditional IAT-CRISPR nucleic acid detection methods, ICB-LAMP-CRISPR/Cas12a has several advantages and provides a single tube detection system. First, in the ICB-LAMP-CRISPR/Cas12a system, ICB is used for sample pretreatment to enrich free pathogens, which greatly improves the sensitivity and specificity of detection. Second, LAMP-CRISPR/Cas12a occurs in a single tube and is mixed by inversion, thus avoiding aerosol pollution. By cutting fluorescent probes with Cas12a, the detection results can be visually analyzed, which avoids the need to display results by lateral flow strips or other methods [48]. When LAMP-CRISPR/Cas12a was used in actual fecal sample detection, the LAMP amplification reagent was prepackaged into the bottom of the PCR tube. Thus, only the sample suspension was prepared, the CRISPR/Cas12a digestion system was added to the lid of the PCR tube, and the steps for rapid visual detection were followed.

The ICB-LAMP-CRISPR/Cas12a assay developed in this study was characterized by high sensitivity and specificity, a short detection time, and nondependent instruments. However, this method can be further improved for more applications in POC. For example, the ICB-LAMP-CRISPR/Cas12a assay could introduce reverse transcriptase into the LAMP system for the detection of RNA pathogens [49]. More conservative target genes or sequences can be obtained by rigorous genome alignment for pathogen detection [50]. The detection objects can be enriched by changing the types of primers used. Integrating recombinase polymerase amplification with the Cas12a enzyme at a common working temperature and realizing a real one-tube detection mixture could completely circumvent the separate preamplification of target nucleic acids and the separation of the Cas enzyme [51]. The LAMP and CRISPR/Cas12a reagents in the ICB-LAMP-CRISPR/Cas12a assay could be lyophilized and stored at room temperature, reducing the cost of cold preservation and transportation of reagents [52]. Alternatively, this ICB-LAMP-CRISPR/Cas12a method could be integrated into a microfluidic chip to enable full integration, sample to result, and multiplexed detection [53,54]. This microfluidic chip, combined with a lyophilized process, can maximize the detection throughput by pre-embedding detection reagents, and the visual detection results can be digitized by mobile phone photography [55,56]. Even, by introducing different fluorescent probes, multiple pathogens in the same reaction vessel can be detected, which would reduce the cost and time for detecting cooccurring pathogens [57]. Furthermore, digital fluorescence signals can promote the application of CRISPR/Cas technology in quantitative detection.

## 5. Conclusions

In summary, an ICB-LAMP-CRISPR/Cas12a detection strategy that can ultrasensitively and timely detect *C. jejuni* in fecal samples based on ICB, LAMP, and the trans-cleavage activity of CRISPR/Cas12a was established. The *C. jejuni* polyclonal antibody-coated ICB was first built for the specific capture of *C. jejuni*. At the same time, the *hipO* (*hippurate hydrolase*) gene (NC_002163.1:c919731-918580) of *C. jejuni* was used as the identification object, and a set of high-efficiency LAMP systems was optimized and obtained. Based on this, CRISPR/Cas12a was designed and introduced to recognize LAMP amplification products. The results showed that the combination of ICB, LAMP, and the CRISPR/Cas12a system can significantly improve the sensitivity and specificity detection of *C. jejuni*. The results could be observed directly under LED blue light. This simple and robust method has potential for the future development of next-generation POC molecular diagnostic technology for the rapid detection of infectious diseases at home or in the breeding industry or clinics.

## Figures and Tables

**Figure 1 biosensors-12-00154-f001:**
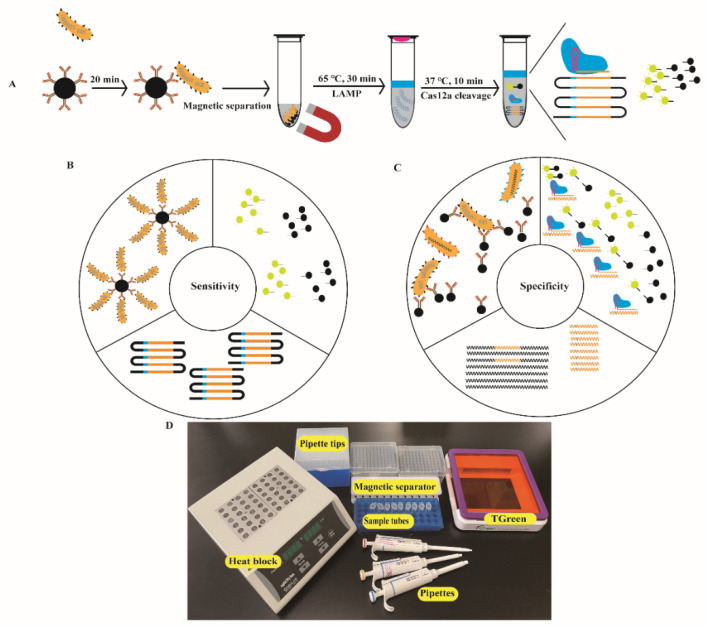
Design and working principle of the ICB-LAMP-CRISPR/Cas12a method. (**A**) Schematic of the ICB-LAMP-CRISPR/Cas12a method. *Campylobacter jejuni* was captured by the prepared ICB and separated magnetically. Five microliters of template DNA of *C. jejuni* was added to the LAMP mixture, which was placed at the bottom of the tube and sealed with 20 μL of mineral oil. The CRISPR/Cas12a reaction reagents are added inside the lid. After 30 min of LAMP amplification at 65 °C, the tube was shaken to mix with Cas12a reagents for cleavage. Once the Cas12a nuclease is activated by recognizing the DNA target, it splits the quenched fluorescent ssDNA-FQ probe indiscriminately, generating a fluorescence signal visible to the naked eye under blue light. (**B**) Enhanced sensitivity of the ICB-LAMP-CRISPR/Cas12a method. The sensitivity was enhanced in three parts: the enrichment of ICB, the high efficiency of LAMP amplification, and the indiscriminate cleavage of the fluorescent ssDNA-FQ probe. (**C**) Enhanced specificity of the ICB-LAMP-CRISPR/Cas12a method. The specificity was enhanced from three parts: the specific antibodies of *C. jejuni* coated in the magnetic beads, the LAMP primers designed based on the conserved *hipO* gene, and the cleavage activity of Cas12a guide by the specific sgRNA. (**D**) Work conditions of the nearly instrument-free POC diagnostics. Equipment and consumables needed for running the ICB-LAMP-CRISPR/Cas12a method include a heat block, pipettes, pipette tips, sample tubes, and T-green transilluminator.

**Figure 2 biosensors-12-00154-f002:**
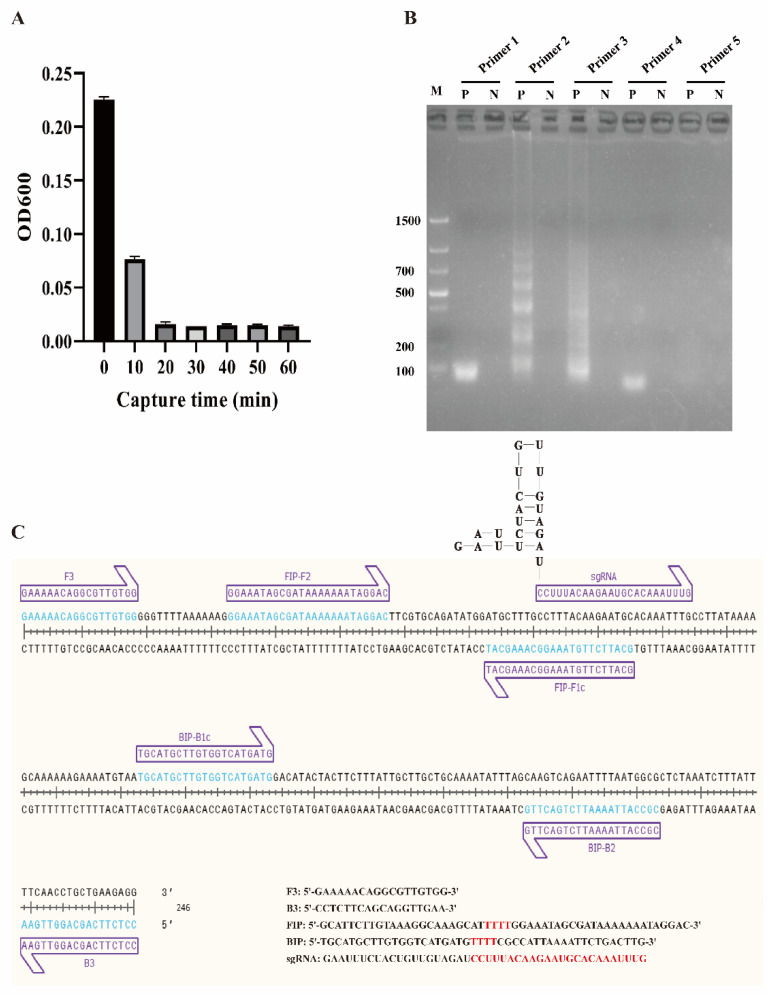
Construction of the ICB-LAMP reaction system. (**A**) Capture time evaluation of ICB. One milliliter of *C. jejuni* (8 × 10^3^ CFU/mL) was mixed with 10 μL of ICB, and the optical density (OD) in the supernatant was measured every 10 min. No significant difference was observed after 20 min of incubation. (**B**) LAMP primer selection. Five groups of LAMP primers based on the *hipO* gene were obtained and used in the LAMP reaction, and primer 2 had the best amplification effect. (**C**) LAMP primers and sgRNA binding sites.

**Figure 3 biosensors-12-00154-f003:**
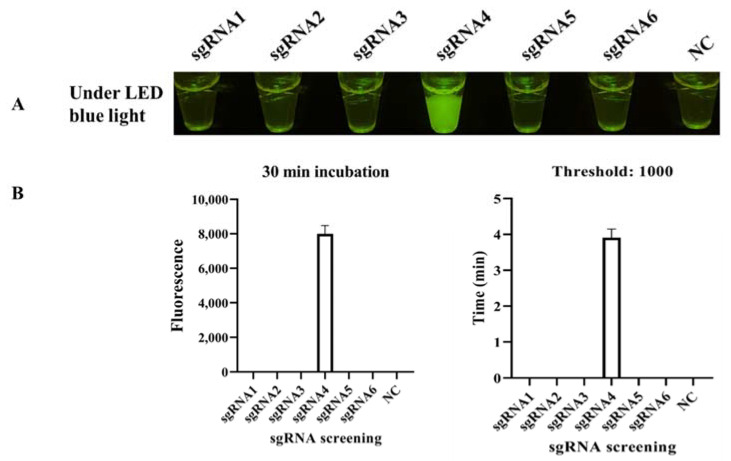
sgRNA screening. (**A**) Endpoint CRISPR/Cas12a for sgRNA screening. Six sgRNAs were designed based on the target DNA and used in CRISPR/Cas12a at 37 °C for 30 min, and sgRNA 4 worked well in the standard program, with strong fluorescent light under LED blue light. (**B**) Real-time CRISPR/Cas12a for sgRNA screening. The real-time monitoring was conducted in a CFX96 Touch Real-Time PCR Detection System for 30 min, and the endpoint fluorescence intensity and time with a fluorescence threshold of 1000 were monitored. Each experiment was repeated three times with similar results.

**Figure 4 biosensors-12-00154-f004:**
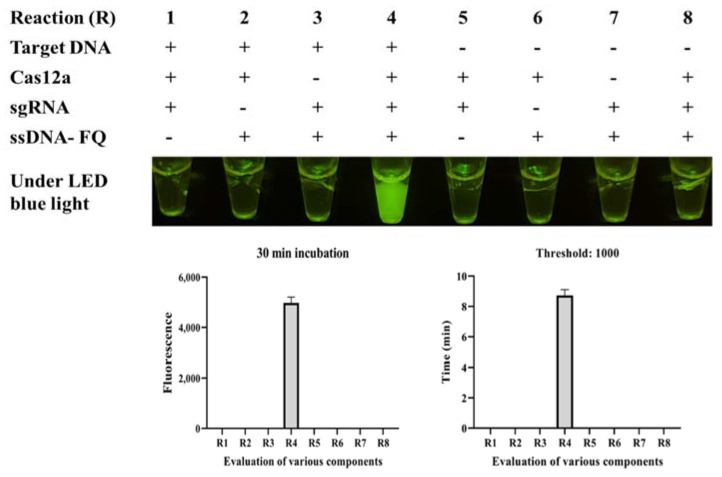
Evaluation of eight CRISPR/Cas12a reactions (R) with various components through endpoint imaging after 30 min of incubation and real-time fluorescence detection. The LAMP products, Cas12a, sgRNA, and the ssDNA-FQ reporter were tested. After incubation at 37 °C for 30 min, only reaction 4, containing the target nucleic acid sequence, sgRNA, Cas12a, and the ssDNA-FQ reporter, produced a superbright fluorescence signal under LED blue light. The real-time monitoring was conducted in a CFX96 Touch Real-Time PCR Detection System for 30 min, and the endpoint fluorescence intensity and time with a fluorescence threshold of 1000 were monitored.

**Figure 5 biosensors-12-00154-f005:**
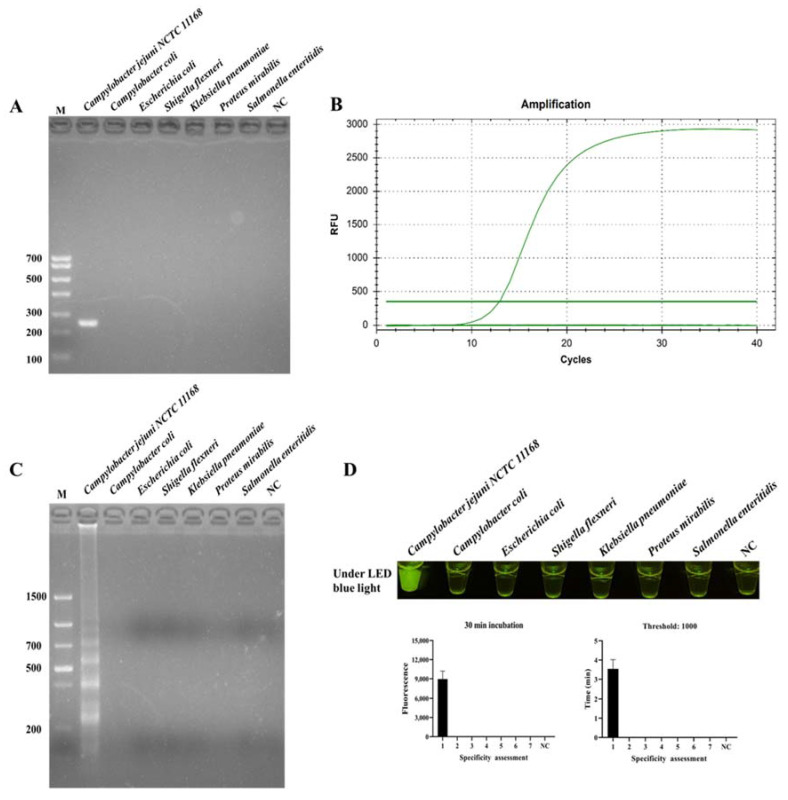
Specificity evaluation ICB-LAMP-CRISPR/Cas12a method. The specificity of ICB-LAMP-CRISPR/Cas12a was compared with PCR, qPCR, and LAMP. *C. jejuni* NCTC 11168 and 6 non-*C. jejuni* (*C. coli*, *E. coli*, *S. flexneri*, *K. pneumoniae*, *P. mirabilis*, and *S. enteritidis*) were used as the tested sample. The specificity evaluation results of PCR and LAMP were shown by agarose gel electrophoresis. The specificity evaluation results of qPCR and ICB-LAMP-CRISPR/Cas12a are shown by the real-time amplification curve and end-point fluorescence. (**A**) The specificity evaluation of PCR. (**B**) The specificity evaluation of qPCR. (**C**) The specificity evaluation of LAMP. (**D**) The specificity evaluation of the ICB-LAMP-CRISPR/Cas12a method.

**Figure 6 biosensors-12-00154-f006:**
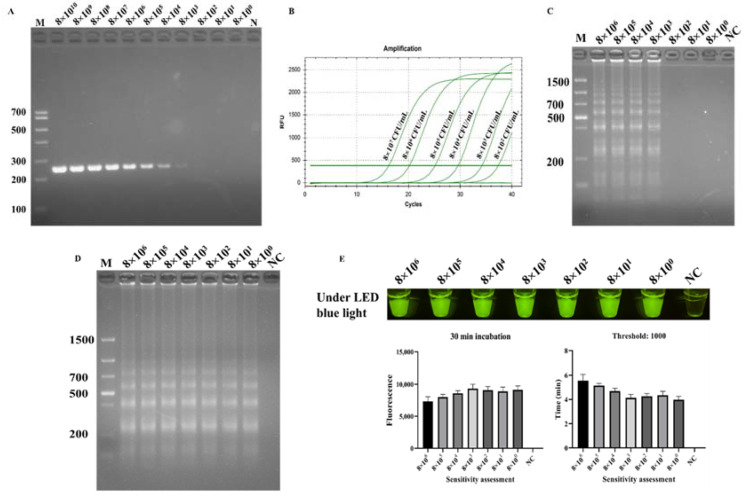
Sensitivity evaluation ICB-LAMP-CRISPR/Cas12a. The sensitivity of ICB-LAMP-CRISPR/Cas12a was compared with- that of PCR, qPCR, LAMP, and ICB-LAMP. Serial tenfold dilutions of *C. jejuni* (8 × 10^0^–8 × 10^10^ CFU/mL) were used for the sensitivity evaluation. The sensitivity evaluation results of PCR, LAMP, and ICB-LAMP were shown by agarose gel electrophoresis. The sensitivity evaluation results of qPCR and ICB-LAMP-CRISPR/Cas12a are shown by the real-time amplification curve and end-point fluorescence. (**A**) The sensitivity evaluation of PCR. (**B**) The sensitivity evaluation of qPCR. (**C**) The sensitivity evaluation of LAMP. (**D**) The sensitivity evaluation of ICB-LAMP. (**E**) The sensitivity evaluation of ICB-LAMP-CRISPR/Cas12a.

**Figure 7 biosensors-12-00154-f007:**
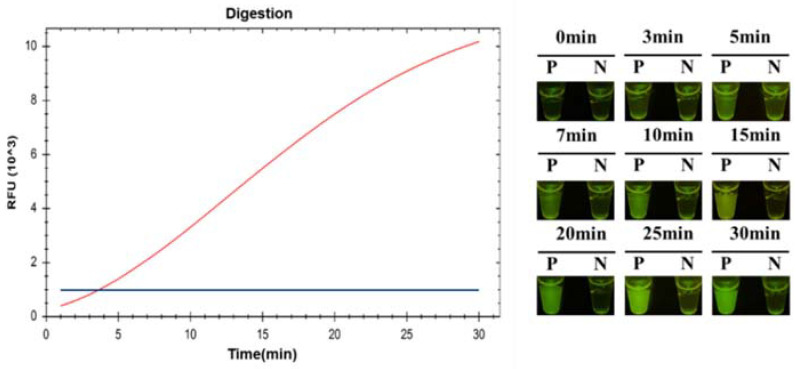
The time evaluation ICB-LAMP-CRISPR/Cas12a. A total of 1 mL 8 × 10^3^ CFU/mL *C. jejuni* was detected by the ICB-LAMP-CRISPR/Cas12a method, and monitored by a CFX96 Touch Real-Time PCR Detection System and real-time photograph detection for 30 min.

**Figure 8 biosensors-12-00154-f008:**
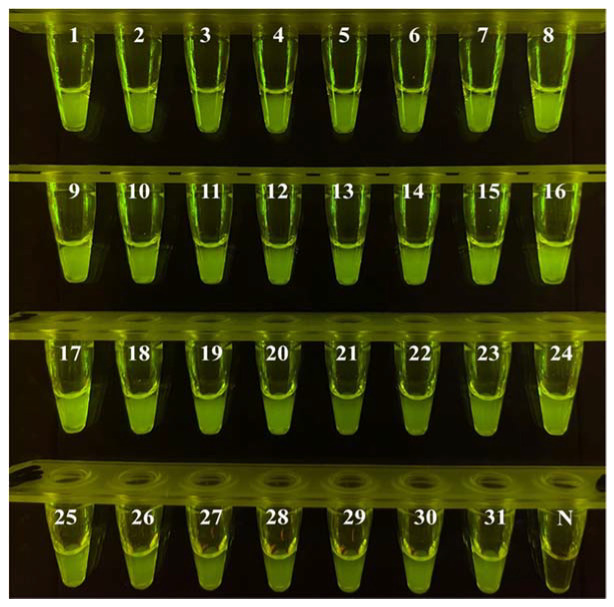
Evaluation of ICB-LAMP-CRISPR/Cas12a in *C. jejuni* isolates. Thirty-one *C. jejuni*-positive fecal samples (n = 31) were used in the actual sample evaluation of ICB-LAMP-CRISPR/Cas12a.

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
