# Peer review of "Immunocapture Magnetic Beads Enhanced the LAMP-CRISPR/Cas12a Method for the Sensitive, Specific, and Visual Detection of Campylobacter jejuni"

_biosensors, 2022, doi:10.3390/bios12030154_

Round 1

Reviewer 1 Report

In manuscript "Pmmunocapture-magnetic beads enhanced LAMP-CRISPR/Cas12a method for the sensitive, specific, and visual detection of Campylobacter jejuni" authors report an all-in-one, simple, sensitive and specific method for the detection of C. jejuni, a food borne bacterial pathogen, which causes gastroenteritis in humans.

The experimental design is appropriate for the described method while the data are presented with the appropriate figures, images and tables.

The results presented in the manuscript are clear and reproducible based on the details given by the authors.

In conclusion, the method presented in this study will greatly benefit the quick identification of the bacterium, as the classical methods require a lot of time ant the sequencing techniques (NGS) need experienced personnel and they are expensive.

Author Response

Response: Thank you very much for your kind help in the evaluation of our manuscript.

Reviewer 2 Report

I read the manuscript submitted by Chao et al with interest. The method may provide a fast diagnostic to detect Campylobacter. But I have some questions for clarification.

1. There is a recent publication about CRISPR-Cas12b based method to detect Campylobacter

Huang Y, Gu D, Xue H, et al. Rapid and Accurate Campylobacter jejuni Detection With CRISPR-Cas12b Based on Newly Identified Campylobacter jejuni-Specific and -Conserved Genomic Signatures. Front Microbiol. 2021;12:649010. Published 2021 Apr 27. doi:10.3389/fmicb.2021.649010

Pls discuss strength and uniqueness compared to this publication. The only major difference I can tell is the combination of ICB with CRISPR-based detection.

2. Campylobacter contamination of poultry is very common, and detecting Campylobacter in poultry cannot say many things. This is a different story from E. coli O157:H7 detection in beef. Because of this, quantitative measurement is more needed than qualitative detection for Campylobacter to assess the level of contamination. However, Figure 6E shows that the fluorescence signal was reduced when CFU exceeded 8x10^3. First, pls explain how this detection pattern in reverse proportion to high CFUs can occur, and discuss if this method can provide a quantitative result.

3. Figure 8 shows the results to validate detection efficacy in fecal samples using 31 Campylobacter-positive fecal samples. I was wondering if the authors tested with Campylobacter-negative fecal samples because there can be non-specific interaction with fecal materials. Alternatively, authors may consider providing data possibly by spiking Campylobacter to Campylobacter-free fecal samples to see if the method can produce any false positive or negative data.

4. Fig 3: Each sgRNA shows very different results. Please explain how sgRNA can be designed and what factors may determine the successful design.

Minor

Figure 1: the figure is too small to read. Also, Fig 1D may not be necessary.

Figure panel labels are small in the entire manuscript.

Author Response

  1. There is a recent publication about CRISPR-Cas12b based method to detect Campylobacter. Huang Y, Gu D, Xue H, et al. Rapid and Accurate Campylobacter jejuni Detection With CRISPR-Cas12b Based on Newly Identified Campylobacter jejuni-Specific and -Conserved Genomic Signatures. Front Microbiol. 2021; 12:649010. Published 2021 Apr 27. doi:10.3389/fmicb.2021.649010. Pls discuss strength and uniqueness compared to this publication. The only major difference I can tell is the combination of ICB with CRISPR-based detection.

Response: Thank you very much for this good article worthy of deep learning, and we read this article carefully. Frankly speaking, these are two types of articles. And at the same time, there is never a perfect technology, only a more suitable application. Professor Jiao's team found a good target in the genome of C. jejuni for rapid detection. This is important and meaningful work. However, in our team, we have jumped out of separate nucleic acid testing and integrated ICB into the nucleic acid test. On the one hand, immunomagnetic beads can specifically identify Campylobacter jejuni in the sample. On the other hand, immunomagnetic beads enrich Campylobacter jejuni into a smaller volume of the aqueous solution, which is equivalent to increasing the concentration of the bacteria and avoiding the interference of other impurities in the sample on amplification and enzyme digestion. From these two perspectives, the specificity and sensitivity of detection are improved compared with the single use of CRISPR, which may be the direction with more development potential. In this paper, the method we adopted is more practical. Of course, if we can in the follow-up work, combined with Professor Jiao's conservative sequence screening technology, will make our work to a higher level. This is quite exciting for us.

  1. Campylobacter contamination of poultry is very common, and detecting Campylobacter in poultry cannot say many things. This is a different story from E. coli O157:H7 detection in beef. Because of this, quantitative measurement is more needed than qualitative detection for Campylobacter to assess the level of contamination. However, Figure 6E shows that the fluorescence signal was reduced when CFU exceeded 8x10^3. First, pls explain how this detection pattern in reverse proportion to high CFUs can occur, and discuss if this method can provide a quantitative result.

Response: CRISPR is a very new technology for in vitro diagnosis, and there are still many unspecified areas, which is the fascinating place of the new technology. Frankly, we cannot figure out the negative correlation (slightly) between fluorescence generation and template concentration for now. We speculated that, first, the assembly of Cas12a and sgRNA was affected due to excessive amplification products, resulting in a slight decrease in the scissors. Second, according to some literature reports (DOI: 10.1038/s41467-020-19072-6), magnesium ions can enhance Cas enzyme activity. Maybe in the constant temperature amplification process, magnesium ions formed precipitation, affecting the Cas enzyme activity. We will try to solve this problem in the follow-up work.

       About the quantitative result by CRISPR/Cas. we think it is very difficult. We know that the output form of CRISPR/Cas results is fluorescence signal. When Cas/sgRNA complex recognizes target DNA, Cas enzyme produces fluorescence through nonspecific cleavage of fluorescent probes. Therefore, the production of fluorescent signals depends on the activity of Cas enzyme and the recognition of target DNA, which are difficult to standardize. Of course, CRISPR/Cas is a very powerful tool. We hope to transform it or tap its more functions to achieve relative or absolute quantification similar to qPCR, which requires a period of effort.

  1. Figure 8 shows the results to validate detection efficacy in fecal samples using 31 Campylobacter-positive fecal samples. I was wondering if the authors tested with Campylobacter-negative fecal samples because there can be non-specific interaction with fecal materials. Alternatively, authors may consider providing data possibly by spiking Campylobacter to Campylobacter-free fecal samples to see if the method can produce any false positive or negative data.

Response: We did not test with Campylobacter-negative fecal samples. About the specificity of this method, we are very confident. As we wrote in the article, the specificity was guaranteed step by step from three aspects. Besides, C. jejuni was captured and isolated from the fecal samples by ICB, and used in the following testing. Even if there are non-C.jejuni substances attached to ICB by electrostatic, specific nucleic acid amplification reaction and specific sgRNA recognition will further strengthen the specificity of the detection.

  1. Fig 3: Each sgRNA shows very different results. Please explain how sgRNA can be designed and what factors may determine the successful design.

Response: The sgRNAs were designed in CHOPCHOP (http://chopchop.cbu.uib.no/). In fact, there is no consensus on the principles of sgRNA design. Even the pioneers of CRISPR/Cas for in vitro diagnostics (Jennifer Doudna and Feng Zhang ) will design hundreds of sgRNA alternatives online from which to test one by one to select usable sgRNAs.

  1. Minor

Figure 1: the figure is too small to read. Also, Fig 1D may not be necessary.

Figure panel labels are small in the entire manuscript.

Response: Thank you very much for your kind help in the evaluation of our manuscript and figure. And about Figure 1D, we think it's a good demonstration of our actual work, indicating that we only need very simple tools to complete the test. So we think it can be retained. As for the font and format of the picture, we will upload clear and qualified large pictures according to the editing requirements.

Reviewer 3 Report

In manuscript "Immunocapture-magnetic beads enhanced
3 LAMP-CRISPR / Cas12a method for the sensitive, specific, and visual detection of Campylobacter jejuni "authors describe the method for the detection of C. jejuni, which integrating immunocapture-magnetic beads (ICB), LAMP, and CRISPR / Cas12a (ICB- LAMP-CRISPR / Cas12a) The problem of proving this bacterium by standard tests is unreliable and we need new methods.
The only thing that is unclear to me is whether they proved the presence of bacteria in a fecal sample or in pure culture? I have no comments on the detailed methodology as well as the presentation of the results. The names of the bacteria should be in italic throughout the text.

Author Response

Response: Thank you very much for your kind help in the evaluation of our manuscript. Yes, we proved the presence of bacteria in a fecal sample, as we wrote in the article, 31 C. jejuni positive fecal samples from different layer farms were stored in our laboratory. Besides, we revised the names of the bacteria to italic throughout the text.